# Distinct Regional Pattern of Sedative Psychotropic Drug Use in South Tyrol: A Comparison with National Trends in Italy

**DOI:** 10.3390/pharmacy13020032

**Published:** 2025-02-21

**Authors:** Christian J. Wiedermann, Katia Sangermano, Pasqualina Marino, Dietmar Ausserhofer, Adolf Engl, Giuliano Piccoliori

**Affiliations:** 1Institute of General Practice and Public Health, College of Healthcare Professions—Claudiana, 39100 Bolzano, Italy; 2Medical Directorate, South Tyrolean Medical Service (SABES-ASDAA), 39100 Bolzano, Italy

**Keywords:** sedative psychotropics, insomnia, benzodiazepines, Z-drugs, sedative antidepressants, melatonin, regional variations, South Tyrol, daily defined dose

## Abstract

This study investigated regional variations in the use of sedative psychotropic medications, often prescribed for insomnia, by comparing Italy and the culturally distinct Autonomous Province of Bolzano, South Tyrol. Using daily defined dose (DDD) data per 1000 inhabitants per day, benzodiazepines, Z-drugs, sedative antidepressants, and melatonin consumption from 2019 to 2023 were examined. The findings indicate a notably lower utilization of benzodiazepines in South Tyrol compared to the national Italian average, alongside a significant increase in sedative antidepressant use, particularly mirtazapine. These disparities likely stem from regional prescribing preferences influenced by cultural, linguistic, and healthcare system factors. While Z-drug consumption remained comparable across regions, melatonin use exhibited a gradual upward trend, albeit less pronounced in South Tyrol. These insights emphasize the necessity for region-specific strategies in optimizing insomnia treatment, balancing pharmacological approaches with non-pharmacological alternatives such as cognitive behavioral therapy for insomnia. Understanding these prescribing trends can inform healthcare policies aimed at reducing long-term sedative use while enhancing patient-centered care in sleep medicine.

## 1. Introduction

Insomnia, a growing global public health issue, is associated with cardiovascular disease, depression, and cognitive impairment [1,2]. Effective management, particularly Cognitive Behavioural Therapy for Insomnia (CBT-I), is becoming increasingly important and recommended as the primary treatment because of its effectiveness and safety [3]. Nevertheless, pharmacological treatments remain prevalent owing to patient preference, accessibility, and limited psychotherapeutic resources [4,5]. Commonly prescribed sedative psychotropic medications include benzodiazepines, non-benzodiazepine hypnotics (Z-drugs), melatonin, and off-label sedative antidepressants [6].

South Tyrol, an autonomous province in northern Italy with a bilingual (German- and Italian-speaking) population, offers a unique setting for studying regional variations in insomnia treatment. Health professionals in both Italian and German countries receive training and continuing education, leading to diverse clinical practices [7,8]. Previous studies have also identified regional differences in health behaviors and healthcare access within Italy, including distinct attitudes toward vaccination and preventive health measures in South Tyrol [9,10]. German-speaking residents in South Tyrol show a notably higher inclination toward complementary and alternative medicine (CAM), indicating potential cultural influences on healthcare choices and preferences [10]. These cultural and systemic factors make South Tyrol a compelling case for exploring regional patterns of psychotropic drug use, particularly sedatives for insomnia.

Previous research has examined psychotropic medication use at a national level in Italy, with studies identifying trends in the general population [11] and providing insights into insomnia-related prescribing patterns [12]. However, these studies do not address sufficiently subnational variations in prescribing behaviors, particularly in bilingual and culturally distinct healthcare settings such as South Tyrol. Comparisons with broader European trends further indicate that benzodiazepine and sedative antidepressant use varies significantly across regions and healthcare systems [13], emphasizing the need for region-specific public health strategies.

Despite the growth of sleep medicine in Italy and the promotion of evidence-based insomnia treatments such as CBT-I, pharmacological approaches remain common [11,12,14]. Digital CBT-I (dCBT-I) can improve access to non-pharmacological treatments, yet sedative psychotropic medication still significantly influences insomnia management, particularly in primary care [12]. However, there is limited data on how sedative psychotropic medication use in South Tyrol aligns with national trends in Italy, especially given the region’s bilingual and culturally unique healthcare system.

This study examined whether South Tyrol’s context affects sedative psychotropic medication use compared to national Italian patterns. By analyzing the daily defined dose (DDD) levels [15] for benzodiazepines, Z-drugs, melatonin, and sedative antidepressants from 2019 to 2023, we investigated whether drug dispensation in South Tyrol aligns with or differs from national trends. To investigate regional prescribing variations, this study aims to address the following research questions:How does the utilization of sedative psychotropic medications in South Tyrol compare to national prescribing trends in Italy from 2019 to 2023?Are there significant differences in the consumption of benzodiazepines, Z-drugs, sedative antidepressants, and melatonin between South Tyrol and Italy?What cultural, healthcare system, and prescribing practice factors contribute to these regional differences?

By addressing these questions, this study seeks to provide a comprehensive understanding of region-specific medication use patterns. The findings may inform future healthcare policies, promoting evidence-based prescribing and adherence strategies tailored to diverse healthcare settings.

## 2. Methods

### 2.1. Study Design and Data Source

This descriptive study is a pharmacoepidemiological analysis of sedative psychotropic medication use in Italy and the Autonomous Province of Bolzano (South Tyrol). A retrospective analysis was conducted on sedative psychotropic medication use (ATC N05 and N06 [16]) for the period from 2019 to 2023. The IQVIA database was used as a data source, which collects the databases of drugs dispensed by the National Health Service (SSN) and purchased privately by the population. Only prescriptions specifically classified under ATC N05 (psycholeptics, including sedatives, hypnotics, and anxiolytics) and N06 (psychoanaleptics, including sedative antidepressants) were included. Data provide information on prescriptions dispensed in WHO-defined daily doses (DDDs) per 1000 inhabitants per day [15], offering a standardized measure of medication use. This approach enables the estimation of the average daily treatment proportion of the population for each drug or drug group. For instance, 10 DDDs per 1000 inhabitants per day suggest that, on average, 1% of the population receives this medication daily.

This dataset does not differentiate between prescriptions from public (SSN) and private providers, nor does it include demographic information on the prescribing population. Consequently, no adjustment for prescriber-specific variations or demographic confounding factors such as age or comorbidities could be implemented.

### 2.2. Medication Grouping

To facilitate targeted analysis, sedative psychotropic medications were categorized based on pharmacological class and typical indications for sleep disturbances, following established classifications such as the WHO Anatomical Therapeutic Chemical (ATC) system [16]. The groups analyzed included benzodiazepines, Z-drugs (non-benzodiazepine hypnotics), sedative antidepressants, and melatonin [17].

#### 2.2.1. Benzodiazepines

This group comprises sedative-anxiolytic medications commonly prescribed for anxiety-related and chronic sleep disturbances, although their use is often limited owing to risk [4]. Benzodiazepines included in this study were alprazolam, bromazepam, brotizolam, clotiazepam, clorazepate, diazepam, etizolam, flunitrazepam, flurazepam, ketazolam, lorazepam, lormetazepam, nitrazepam, nordazepam, oxazepam, pinazepam, prazepam, and triazolam.

#### 2.2.2. Z-Drugs (Non-Benzodiazepine Hypnotics)

Known for their selective action on GABA-A receptors, Z-drugs are typically first-line agents for short-term sleep initiation disorders and are associated with fewer residual daytime effects than traditional benzodiazepines [18]. This group included patients treated with zopiclone, zolpidem, and eszopiclone.

#### 2.2.3. Sedative Antidepressants

Sedative antidepressants are mostly used off-label to treat sleep disturbances, particularly in cases where insomnia co-occurs with depressive symptoms [19]. Medications in this category include mirtazapine, amitriptyline (in combination with antipsychotics), clomipramine, and trimipramine.

#### 2.2.4. Melatonin

As an over-the-counter and prescription supplement, prolonged-release melatonin is used to manage circadian rhythm disorders and mild sleep disturbances, providing a non-habit-forming alternative that is especially suitable for patients with circadian-related sleep issues [20].

This structured categorization, grounded in recognized pharmacological classifications and clinical guidelines, supported the consistency and validity of the analysis.

### 2.3. Data Analysis

For each medication group, we calculated the DDD per 1000 residents per day for each year between 2019 and 2023. Annual means and standard deviations were determined to summarize usage patterns. Comparisons were made between South Tyrol and the broader national averages for Italy, with a focus on identifying regional differences in utilization trends over time. Spearman’s rank correlation analysis was applied to assess trends within each medication group over time (n = 5 years). Sedative psychotropics were ranked by national DDD values to identify differences in the most frequently prescribed compounds between regions. Statistical analyses were conducted using the Jeffreys’ Amazing Statistics Program (JASP; University of Amsterdam, Amsterdam, The Netherlands).

## 3. Results

Sedative psychotropic medication utilization in Italy and South Tyrol from 2019 to 2023 was expressed as DDD per 1000 inhabitants per day across four drug classes: benzodiazepines, Z-drugs, sedative antidepressants, and melatonin. These data provide an annual representation of consumption patterns (Table 1).

Table 1 and Figure 1 show that Italy’s overall consumption of sedative psychotropic drugs exceeded that of South Tyrol, with both regions displaying stable trends. Italy consistently used more benzodiazepines, whereas South Tyrol showed a notable decline in their use over time, possibly indicating changes in prescribing practices or a preference for alternative therapies in the region. Z-drug usage was similar in Italy and South Tyrol, with a rising trend in Italy, indicating increased reliance on these medications instead of benzodiazepines. Sedative antidepressants are more frequently used in South Tyrol, showing a gradual increase, possibly due to regional prescription preferences or a preference for these over traditional anxiolytics. Melatonin usage increased in both regions, although it was marginally lower in South Tyrol than in Italy. This trend suggests a growing preference for non-benzodiazepines and non-sedative sleep support options. Overall, these differences indicate distinct healthcare practices and cultural attitudes toward sedative medications for sleep issues.

In Italy, the seven most frequently used sedative psychotropic drugs account for 80.2% of the total consumption, representing 17.6 of the 21.9 DDD per 1000 inhabitants per day across 28 psychotropics. As shown in Table 2, the results reveal distinct regional differences in consumption patterns and trends over the observation period from 2019 to 2023.

Lormetazepam ranks as the top sedative psychotropic in both areas, with its use in Italy nearly twice that of South Tyrol. Italy’s usage remains steady, whereas South Tyrol shows a marked decline, indicating a regional shift from this benzodiazepine.Alprazolam ranks second, with Italy showing a higher usage than South Tyrol. In Italy, alprazolam use is relatively steady, whereas South Tyrol exhibits a modest, stable pattern with consistently lower DDD values, likely indicating a more conservative prescription approach.Lorazepam is the third most used drug, with higher usage in Italy than in South Tyrol. Italy exhibited a slight decline, whereas South Tyrol showed a significant decrease, indicating a shift in prescribing practices favoring alternatives.Zolpidem, a Z-drug, exhibits similar usage in Italy and South Tyrol, with a slight increase in both areas. Italy’s use trends were marginally higher, showing a comparable acceptance of this non-benzodiazepine hypnotic in both regions.Triazolam usage was slightly higher in Italy than in South Tyrol, where a mild decrease contrasts with Italy’s stable pattern, possibly reflecting a preference for other short-acting hypnotics in South Tyrol.Mirtazapine, a sedative antidepressant, was used at significantly higher levels in South Tyrol than in Italy, with increasing trends in both regions. This indicates that South Tyrol may favor sedative antidepressants over benzodiazepines as alternatives for managing sleep disturbances.Benzodiazepine derivatives (unspecified, including infrequently used drugs) exhibit moderate and stable usage in Italy. In contrast, South Tyrol demonstrated significantly lower and declining usage, indicating reduced dependence on these agents.

These top quarter drugs highlight Italy’s greater reliance on benzodiazepines than South Tyrol.

The results indicated significant regional differences in sedative psychotropic drug use between Italy and South Tyrol. Italy showed higher benzodiazepines, with stable trends, reflecting consistent prescribing practices. Conversely, South Tyrol exhibited a marked decrease in benzodiazepine use, indicating a regional shift toward reducing reliance on these medications. The increase in sedative and antidepressant use, particularly mirtazapine, in South Tyrol suggests a preference for alternatives to benzodiazepines, aligning with the cautious approach of prescribing anxiolytics and hypnotics. Z-drug class utilization is comparable, with zolpidem showing increased use in Italy but not in South Tyrol. The rising trend in melatonin use in both regions reflects the growing acceptance of non-habit-forming sleep aids, although at a lower level in the South Tyrol.

## 4. Discussion

Based on a 2023 consensus update on the management of insomnia in clinical practice in Italy [14], current recommendations emphasize that CBT-I should be the first-line treatment owing to its effectiveness and safety profile, particularly for chronic insomnia. Pharmacological treatments, including sedative psychotropic agents, are generally advised only when CBT-I is unavailable or rapid symptom relief is necessary for transient insomnia. Specifically, benzodiazepines and Z-drugs are recommended only for short-term use (up to four weeks), while the newer Z-drug eszopiclone may be considered for extended use up to six months, particularly in elderly patients. Melatonin and the novel dual orexin receptor antagonist daridorexant have been suggested as options for long-term insomnia management, with specific indications based on patient age and insomnia severity [14].

The results presented here elucidate significant regional disparities in sedative psychotropic drug utilization, with Italy exhibiting higher overall levels compared to South Tyrol, particularly in the use of benzodiazepines (Table 1). The consistent utilization of benzodiazepines in Italy likely reflects a sustained reliance on traditional anxiolytics and hypnotics, whereas South Tyrol demonstrates a notable reduction in this class, indicating a regional shift that may prioritize minimizing benzodiazepine use because of concerns regarding dependency, especially among elderly populations. This trend in South Tyrol aligns with the observed increase in sedative antidepressant use, specifically mirtazapine, a drug off-label for insomnia in Italy [14]. The increase in antidepressant use with sedative properties in South Tyrol may indicate a deliberate choice of medications that offer additional benefits for treating comorbid depression. Nevertheless, the general agreement highlights the lack of substantial evidence supporting the prolonged use of antidepressants to address sleep disorders [14].

Z-drugs, such as zolpidem, demonstrate relatively comparable utilization across both regions, although Italy exhibits a slight upward trend. This comparable usage may indicate a shared acceptance of Z-drugs as a benzodiazepine alternative for short-term insomnia treatment. Eszopiclone, which was recently introduced to the Italian market, attained a DDD of 0.02 per 1000 inhabitants in Italy in 2023. However, it has not yet been introduced in the South Tyrol. This disparity likely reflects the varying local prescribing practices. Regional variations in prescribing patterns suggest potential cultural and systemic influences, but the relationship between Z-drug prescribing and non-pharmacological insomnia treatments requires further investigation. Increased Z-drug use alongside lower benzodiazepine prescriptions may indicate a shift toward alternative pharmacological management rather than direct substitution for CBT-I. While CBT-I is recommended as a first-line treatment for insomnia, its accessibility remains limited in many regions, and patient adherence can be variable. Where CBT-I is unavailable, insufficiently utilized, or ineffective, physicians may resort to Z-drugs as a fallback option.

Melatonin usage, while low, increased across both Italy and South Tyrol, indicating a growing acceptance of non-prescription, non-benzodiazepine options for sleep regulation, particularly in cases of mild insomnia or circadian rhythm disturbances. The relatively lower level in South Tyrol might indicate either limited acceptance or more cautious adoption compared to Italy.

A recent study conducted by the Statistics Institute of the Autonomous Province of Bolzano (ASTAT) and the Institute of General Practice and Public Health Bolzano found that most adults in South Tyrol rate their sleep quality positively. Using the Pittsburgh Sleep Quality Index (short version), this study revealed that 82% of the participants perceived their sleep as fairly or very good, while 18% reported poor sleep quality. The prevalence of sleep difficulties in South Tyrol is influenced by factors such as gender, age, and chronic health conditions, with women, older adults, and those with chronic conditions experiencing more sleep issues [21].

A key finding is the lower prevalence of reported sleep disturbances among the German-speaking majority in South Tyrol compared to the Italian-speaking minority, with German speakers experiencing less frequent sleep quality issues and more consistent adherence to sufficient sleep habits [21]. This observation may partially explain the lower overall use of sedative psychotropics in South Tyrol than in Italy, as a larger portion of South Tyrol’s German-speaking population may experience fewer sleep disruptions, reducing the demand for sedative medications.

This study explores cultural influences on prescribing practices in South Tyrol, primarily considering language groups as a proxy for cultural variation. However, cultural diversity in the region is broader, including Ladin-speaking communities and populations with a migrant background, who may have different healthcare access patterns and preferences. The single-payer healthcare system in Italy generally ensures uniform treatment guidelines, but individual beliefs, health-seeking behaviors, and socioeconomic factors could influence prescribing patterns [22]. Due to data limitations, this study could not analyze healthcare access disparities among specific ethnic subgroups or compare treatment-seeking behaviors across cultural backgrounds. Future research should assess how cultural beliefs, health literacy, and structural access barriers impact insomnia treatment decisions, particularly in multicultural and bilingual healthcare settings.

### 4.1. Implications for Healthcare Providers and Public Health Policy

The findings underscore the necessity for targeted interventions to optimize insomnia treatment and mitigate long-term dependence on sedative psychotropic medications, particularly benzodiazepines [23]. Healthcare practitioners should receive comprehensive training on insomnia management, emphasizing the risks associated with prolonged benzodiazepine use and the benefits of non-pharmacological alternatives such as Cognitive Behavioral Therapy of Insomnia (CBT-I) [24]. Continuing medical education programs should incorporate strategies for deprescribing benzodiazepines and ensuring appropriate patient follow-up.

Enhancing access to CBT-I, particularly through digital platforms (dCBT-I), could address barriers related to availability and cost [25]. Policymakers should consider integrating CBT-I into primary care settings as the first-line treatment for insomnia. Public health authorities could initiate campaigns to educate patients on sleep hygiene, non-pharmacological approaches, and the potential adverse effects of chronic sedative use [26]. Such initiatives could alter perceptions of insomnia management and reduce demand for prescription sedatives.

In regions with high benzodiazepine utilization, prescription monitoring programs could facilitate the tracking of prescribing trends and promote safer practices [27]. Policy measures, such as restricting long-term benzodiazepine prescriptions to specific cases and mandating a stepwise approach incorporating CBT-I prior to sedative use, could align prescribing habits with evidence-based guidelines.

### 4.2. Strengths and Limitations

This study has several limitations. DDD per 1000 inhabitants per day provides a standardized measure for comparing medication use across regions but does not capture individual-level data such as patient demographics, clinical indications, or prescribing physician characteristics. Consequently, this study cannot account for age-specific prescribing patterns, comorbidities, or dosage adjustments based on medical conditions. DDD values do not necessarily reflect actual doses consumed by patients, as adherence and off-label prescribing practices are not captured.

Another limitation is the lack of differentiation between public (SSN) and private prescriptions in the IQVIA database, preventing the assessment of prescriber-specific variations. Differences in healthcare access, private insurance coverage, or physician preferences may have influenced regional trends but could not be explicitly analyzed. This study also lacks data on specific indications for prescription, making it impossible to distinguish whether sedative psychotropics were prescribed primarily for insomnia or other conditions such as anxiety disorders or depression.

Additionally, while South Tyrol provides an interesting case study, the specific findings may not be generalizable to other bilingual or multicultural regions with different healthcare systems. The absence of individual-level comorbidity data limits the clinical applicability of these findings, particularly in patients with co-existing conditions. Furthermore, treatment availability, reimbursement structures, and healthcare policy differences may influence prescribing behaviors but were not explicitly assessed.

While prior studies have noted increased sedative psychotropic use during the COVID-19 pandemic, this dataset only allows for the identification of indicative trends rather than causal relationships. Although some medications exhibited peak DDD levels during the observation period, these fluctuations cannot be definitively attributed to pandemic-related changes in prescribing behavior without further individual-level data.

Notwithstanding these limitations, the utilization of a comprehensive national database ensures reliable comparisons between South Tyrol and the broader Italian healthcare system. Subsequent research should incorporate individual-level data to refine the understanding of prescribing patterns, particularly regarding patient characteristics, adherence behaviors, treatment protocols, and the influence of healthcare policies on sedative psychotropic use.

The strength of this study is its focus on South Tyrol, a unique cross-cultural region influenced by Italian and German-speaking healthcare practices, enabling a comparison of sedative psychotropic drug use in a single system that serves two linguistic and cultural communities. Using DDD as a standardized metric allows the systematic analysis of drug utilization trends over time, facilitating robust comparisons at the national and regional levels. The multi-year observation period provides insights into longitudinal trends, capturing shifts in prescribing practices and the impact of cultural and regulatory differences on medication choice, offering a better understanding of regional variations in insomnia management and highlighting opportunities for evidence-based, non-pharmacological treatments.

### 4.3. Future Research Directions

While this study provides insights into regional prescribing patterns of sedative psychotropic medications, further research is needed to understand the underlying factors driving these trends. Future studies should examine how age, comorbidities, and adherence behaviors influence medication use. Investigating the effectiveness of deprescribing initiatives and alternative insomnia treatments, such as CBT-I implementation in primary care, would be essential for refining clinical guidelines [28]. Comparative studies across other culturally distinct regions could help assess whether regional prescribing trends in South Tyrol reflect broader healthcare system differences or unique sociocultural factors. Longitudinal analyses could provide insights into the long-term impact of shifting prescribing practices on patient outcomes and healthcare resource utilization.

## 5. Conclusions

This study revealed significant regional differences in sedative psychotropic drug use, with Italy demonstrating higher overall levels, particularly in benzodiazepine use, than South Tyrol. The reduced benzodiazepine uses and increased utilization of sedative antidepressants, notably mirtazapine, in South Tyrol suggests a shift towards alternatives considered more appropriate for long-term use. Z-drug use was comparable in both regions. Rising melatonin use reflects an increased acceptance of non-prescription options for mild sleep disturbances, although melatonin uptake remains lower in South Tyrol. The limitations of DDD data, including the absence of individual-level and indication-specific insights, restrict more detailed conclusions. Nonetheless, this study underscores the cultural and regulatory impact of sedative drug use in South Tyrol. Future research should incorporate individual-level data to better elucidate prescribing motivations and assess the role of non-pharmacological treatments in reducing sedative psychotropic use.

## Figures and Tables

**Figure 1 pharmacy-13-00032-f001:**
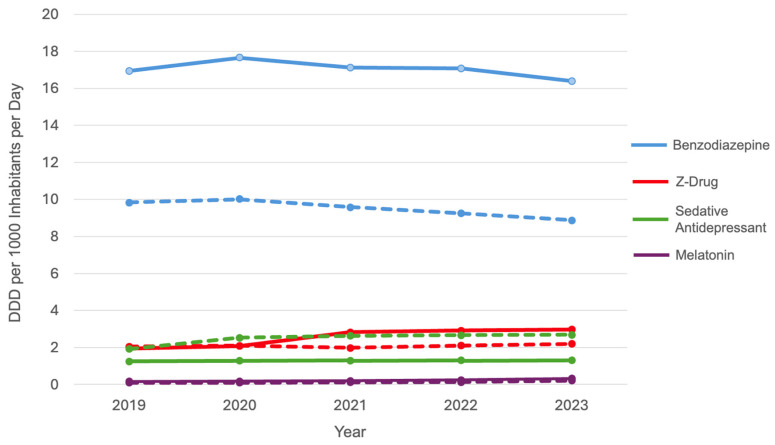
Trends in sedative psychotropic drug use in Italy and South Tyrol (2019–2023). Solid lines represent Italy, and dashed lines represent South Tyrol. DDD, defined daily dose.

**Table 1 pharmacy-13-00032-t001:** Provision patterns of sedative psychotropic drugs in Italy and South Tyrol in daily defined dose (DDD) per 1000 inhabitants with correlation analysis and regional comparison from 2019 to 2023.

Region	Sedative Psychotropic Drug ^2^	DDD per 1000 Inhabitants per Day ^1^									
		Mean ^3^	SD ^3^	*p*-Value ^4^	2019	2020	2021	2022	2023	r_s_ ^5^	*p*-Value ^6^
Italy	Benzodiazepines	17.04	0.454	0.0079	16.94	17.66	17.12	17.07	16.39	0.4000	n.s.
	Z-Drugs	2.55	0.495	n.s.	1.94	2.08	2.82	2.92	2.97	1.0000	0.0200
	Sedative Antidepressants	1.29	0.020	0.0079	1.25	1.28	1.29	1.30	1.31	1.0000	0.0200
	Melatonin	0.22	0.063	n.s.	0.16	0.18	0.2	0.24	0.32	1.0000	0.0200
	Total	21.09	0.493	0.0079	20.29	21.20	21.43	21.53	20.99	0.4000	n.s.
South Tyrol	Benzodiazepines	9.50	0.455	---	9.84	10.01	9.58	9.25	8.88	−0.9000	0.0374
	Z-Drugs	2.09	0.080	---	2.05	2.09	1.98	2.11	2.20	0.7000	n.s.
	Sedative Antidepressants	2.62	0.322	---	1.92	2.53	2.63	2.66	2.69	0.8198	0.05
	Melatonin	0.13	0.049	---	0.09	0.09	0.12	0.13	0.21	0.9750	0.0048
	Total	14.34	0.325	---	13.9	14.72	14.31	14.15	13.98	0.0000	n.s

^1^ Rank by mean DDD per 1000 inhabitants per day in Italy from 2019 to 2024. ^2^ Benzodiazepines (alprazolam, benzodiazepine derivatives, bromazepam, brotizolam, clotiazepam, clorazepate, diazepam, etizolam, flunitrazepam, flurazepam, ketazolam, lorazepam, lormetazepam, nitrazepam, nordazepam, oxazepam, pinazepam, prazepam, triazolam); Z-drugs (zopiclone, zolpidem, eszopiclone); sedative antidepressants (mirtazapine, amitriptyline, amitriptyline combinations with antipsychotics, clomipramine, trimipramine). ^3^ Mean and standard deviation of DDD per 1000 inhabitants per day from 2019 to 2024. ^4^ Mann–Whitney U-Test of DDD values Italy against South Tyrol. ^5^ Spearman’s rank correlation coefficient between year and DDD per 1000 inhabitants per day. ^6^ Spearman’s rank correlation. Abbreviations: DDD, defined daily dose; n.s., not significant (*p* ≥ 0.05).

**Table 2 pharmacy-13-00032-t002:** Utilization patterns of sedative psychotropic drugs in Italy and South Tyrol (DDD per 1000 inhabitants per day), with correlation analysis and regional comparison (2019–2023).

Rank ^1^	Sedative Psychotropic Drug	Region	DDD per 1000 Inhabitants per Day									
			Mean ^2^	SD	*p*-Value ^3^	2019	2020	2021	2022	2023	r_s_ ^4^	*p*-Value ^5^
1	Lormetazepam	Italy	5.05	0.091	0.0119	4.98	5.14	5.06	5.14	4.94	−0.2052	n.s.
		South Tyrol	2.67	0.167		2.83	2.80	2.73	2.56	2.44	−1.0	<0.0001
2	Alprazolam	Italy	3.79	0.12	0.0079	3.60	3.90	3.83	3.87	3.75	0.1	n.s.
		South Tyrol	1.2	0.062		1.10	1.25	1.24	1.22	1.17	0	<0.0001
3	Lorazepam	Italy	3.73	0.184	0.0119	3.83	3.96	3.74	3.63	3.48	−0.9	0.0374
		South Tyrol	2.44	0.143		2.58	2.58	2.44	2.36	2.25	−0.9747	0.0048
4	Zoldipem	Italy	2.05	0.133	n.s.	1.86	2.00	2.05	2.15	2.20	1.0	<0.0001
		South Tyrol	2.06	0.081		2.02	2.06	1.95	2.08	2.17	0.7	n.s.
5	Triazolam	Italy	1.35	0.029	0.0157	1.40	1.40	1.35	1.35	1.32	−0.3591	n.s.
		South Tyrol	1.22	0.072		1.26	1.32	1.21	1.17	1.14	−0.9	0.0374
6	Mirtazapin	Italy	0.67	0.026	0.0079	0.64	0.66	0.67	0.68	0.71	1.0	<0.0001
		South Tyrol	1.9	0.076		1.18	1.84	1.91	1.94	2.00	1.0	<0.0001
7	Benzodiazepine (Derivatives)	Italy	0.93	0.025	0.0095	0.89	0.95	0.94	0.94	0.91	0.0513	n.s.
		South Tyrol	0.54	0.022		0.53	0.58	0.53	0.53	0.53	−0.3536	n.s.
8	Brotizolam	Italy	0.53	0.012	n.s.	0.54	0.54	0.53	0.53	0.51	−0.9487	0.0138
		South Tyrol	0.51	0.035		0.57	0.52	0.51	0.49	0.48	−1.0	<0.0001
9	Bromazepam	Italy	0.52	0.038	0.0117	0.55	0.55	0.52	0.50	0.46	−0.9747	0.0048
		South Tyrol	0.21	0.013		0.22	0.22	0.21	0.20	0.19	−0.9747	0.0048
10	Zopiclone	Italy	0.49	0.374	n.s.	0.08	0.08	0.77	0.77	0.75	0.6325	n.s.
		South Tyrol	0.03	0		0.03	0.03	0.03	0.03	0.03	n.a.	n.a.
11	Diazepam	Italy	0.48	0.018	0.0109	0.48	0.51	0.49	0.48	0.46	−0.5643	n.s.
		South Tyrol	0.14	0.011		0.14	0.14	0.12	0.15	0.14	0.2236	n.s.
12	Amitryptiline	Italy	0.31	0.005	0.0107	0.31	0.30	0.30	0.31	0.31	0.2887	n.s.
		South Tyrol	0.52	0.011		0.53	0.50	0.51	0.52	0.52	−0.0513	n.s.
13	Flurazepam	Italy	0.25	0.004	0.0088	0.25	0.25	0.25	0.25	0.24	−0.7071	n.s.
		South Tyrol	0.21	0.009		0.20	0.22	0.22	0.22	0.21	0.2236	n.s.
14	Melatonin	Italy	0.22	0.063	n.s.	0.16	0.18	0.20	0.24	0.32	1.0	<0.0001
		South Tyrol	0.13	0.049		0.09	0.09	0.12	0.13	0.21	0.9747	0.0048
15	Clomipramine	Italy	0.21	0.005	0.0099	0.20	0.21	0.21	0.21	0.20	0	n.s.
		South Tyrol	0.17	0.014		0.18	0.16	0.18	0.18	0.15	−0.4472	n.s.
16	Prazepam	Italy	0.11	0.038	0.0107	0.14	0.14	0.13	0.10	0.05	−0.9747	0.0048
		South Tyrol	0.03	0.011		0.04	0.04	0.04	0.02	0.02	−0.866	n.s.
17	Etizolam	Italy	0.12	0	0.0065	0.12	0.12	0.12	0.12	0.12	n.a.	n.a.
		South Tyrol	0.07	0.005		0.08	0.08	0.07	0.07	0.07	−0.866	n.s.
18	Amitriptyline (Combination with Antipsychotic Agent)	Italy	0.1	0.004	0.0086	0.10	0.10	0.10	0.10	0.09	−0.7071	n.s.
		South Tyrol	0.03	0.005		0.03	0.03	0.03	0.02	0.02	−0.866	n.s.
19	Clotiazepam	Italy	0.06	0.018	0.0088	0.09	0.07	0.05	0.05	0.05	−0.8944	0.0405
		South Tyrol	0.02	0.004		0.03	0.02	0.02	0.02	0.02	−0.7071	n.s.
20	Flunitrazepam	Italy	0.03	0.008	0.0107	0.04	0.04	0.03	0.03	0.02	−0.9487	0.0138
		South Tyrol	0.06	0.007		0.07	0.06	0.06	0.06	0.05	−0.8944	0.0405
21	Oxazepam	Italy	0.03	0	0.004	0.03	0.03	0.03	0.03	0.03	n.a.	n.a.
		South Tyrol	0.13	0		0.13	0.13	0.13	0.13	0.13	n.a.	n.a.
22	Ketazolam	Italy	0.02	0	0.004	0.02	0.02	0.02	0.02	0.02	n.a.	n.a.
		South Tyrol	0	0		0	0	0	0	0	n.a.	n.a.
23	Nitrazepam	Italy	0.02	0	0.0056	0.02	0.02	0.02	0.02	0.02	n.a.	n.a.
		South Tyrol	0.03	0.004		0.04	0.03	0.03	0.03	0.03	−0.7071	n.s.
24	Clorazepate	Italy	0.01	0	0.177	0.01	0.01	0.01	0.01	0.01	n.a.	n.s.
		South Tyrol	0.01	0.005		0.02	0.02	0.01	0.01	0.01	−0.8660	n.a.
25	Trimipramine	Italy	0	0.005	n.s.	0	0.01	0.01	0	0	−0.2887	n.s.
		South Tyrol	0	0		0	0	0	0	0	n.a.	n.a.
26	Nordazepam	Italy	0	0	n.s.	0	0	0	0	0	n.a.	n.a.
		South Tyrol	0	0		0	0	0	0	0	n.a.	n.a.
27	Pinazepam	Italy	0	0	0.02	0	0	0	0	0	n.a.	n.a.
		South Tyrol	0.01	0.004		0.01	0.01	0.01	0.01	0	−0.7071	n.s.
28	Eszopiclone	Italy	0	0.009	n.s.	0	0	0	0	0.02	0.7071	n.s.
		South Tyrol	0	0		0	0	0	0	0	n.a.	n.a.

^1^ Rank by mean DDD per 1000 inhabitants per day in Italy from 2019 to 2024. ^2^ Mean and standard deviation of DDD per 1000 inhabitants per day in Italy from 2019 to 2024. ^3^ Mann–Whitney U-Test of DDD values Italy against South Tyrol. ^4^ Spearman’s rank correlation coefficient between year and DDD per 1000 inhabitants per day for trend analysis. ^5^
*p*-value corresponding to Spearman’s rank correlation coefficient. Abbreviations: DDD, defined daily dose; n.a., not analyzed; n.s., not significant (*p* ≥ 0.05).

## Data Availability

Data are available from the corresponding author upon reasonable request.

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
