# Peer review of "Distinct Regional Pattern of Sedative Psychotropic Drug Use in South Tyrol: A Comparison with National Trends in Italy"

_pharmacy, 2025, doi:10.3390/pharmacy13020032_

Round 1

Reviewer 1 Report

Comments and Suggestions for Authors

The abstract could significantly enhance its impact by more passionately underscoring the study's significance and the potential implications it holds.

Enhance the introduction by clearly delineating the specific research questions or hypotheses guiding the study.

Consider expanding the methodological section to include more detailed descriptions of data processing, handling of potential confounders, and justification for chosen statistical methods.

 The discussion highlights the relevance of cognitive-behavioural therapy for insomnia (CBT-I) and the shift away from benzodiazepines. To make the paper more actionable, consider elaborating on how healthcare providers, policymakers, or public health authorities might use these findings to inform targeted educational programs or policy measures, particularly in regions where sedative drug usage remains high.

Provide a more comprehensive discussion of the study's limitations.

Elaborate on the practical implications of the findings for healthcare providers and policymakers. Additionally, outline specific avenues for future research to build upon the study's findings.

Comments on the Quality of English Language

Minor grammatical issues are present.

Author Response

Comment 1: The abstract could significantly enhance its impact by more passionately underscoring the study's significance and the potential implications it holds.

Response 1: We have revised it to emphasize the broader significance of our findings and their implications for optimizing medication use and adherence in insomnia management. The updated abstract highlights the relevance of cultural and healthcare system influences on prescribing practices and the potential for non-pharmacological interventions.

Comment 2: Enhance the introduction by clearly delineating the specific research questions or hypotheses guiding the study.

Response 2: We have added a new paragraph at the end of the Introduction section, explicitly stating the key research questions.

Comment 3: Consider expanding the methodological section to include more detailed descriptions of data processing, handling of potential confounders, and justification for chosen statistical methods.

Response 3: In response to the reviewer's suggestions, the Methods section was revised to enhance clarity and provide additional methodological details: Clarification on Data Limitations - A statement was added to Section 2.1 (Study Design and Data Source) specifying that the dataset does not differentiate between public (SSN) and private prescriptions and lacks demographic details on the prescribing population. Improved Description of Data Analysis - In Section 2.3 (Data Analysis), additional detail was included to specify that annual means and standard deviations were calculated to summarize medication usage patterns, making the statistical approach clearer. The statistical software used (Jeffreys’ Amazing Statistics Program (JASP)) was explicitly mentioned to enhance reproducibility.

Comment 4: The discussion highlights the relevance of cognitive-behavioural therapy for insomnia (CBT-I) and the shift away from benzodiazepines. To make the paper more actionable, consider elaborating on how healthcare providers, policymakers, or public health authorities might use these findings to inform targeted educational programs or policy measures, particularly in regions where sedative drug usage remains high.

Response 4: The following subchapter has been inserted in the Discussion section in response:
4.1. Implications for Healthcare Providers and Public Health Policy
The findings underscore the necessity for targeted interventions to optimise insomnia treatment and mitigate long-term dependence on sedative psychotropic medications, particularly benzodiazepines [21]. Healthcare practitioners should receive comprehensive training on insomnia management, emphasising the risks associated with prolonged benzodiazepine use and the benefits of non-pharmacological alternatives such as Cognitive Bahavioral Therapy of Insomnia (CBT-I) [22]. Continuing medical education programmes should incorporate strategies for deprescribing benzodiazepines and ensuring appropriate patient follow-up.
Enhancing access to CBT-I, particularly through digital platforms (dCBT-I), could address barriers related to availability and cost [23]. Policymakers should consider integrating CBT-I into primary care settings as the first-line treatment for insomnia. Public health authorities could initiate campaigns to educate patients on sleep hygiene, non-pharmacological approaches, and potential adverse effects of chronic sedative use [24]. Such initiatives could alter perceptions of insomnia management and reduce demand for prescription sedatives.
In regions with high benzodiazepine utilisation, prescription monitoring programmes could facilitate the tracking of prescribing trends and promote safer practices [25]. Policy measures, such as restricting long-term benzodiazepine prescriptions to specific cases and mandating a stepwise approach incorporating CBT-I prior to sedative use, could align prescribing habits with evidence-based guidelines.

Comment 5: Provide a more comprehensive discussion of the study's limitations.

Response 5: We have expanded this section to clarify the constraints of using DDD as a measurement tool, including its inability to capture individual-level data, prescriber-specific variations, and clinical indications for medication use. We have also incorporated a discussion on the lack of differentiation between public and private prescriptions, the inability to assess adherence behaviors, and the limitations in attributing prescribing trends to the COVID-19 pandemic. These revisions enhance the transparency of our findings and provide a clearer context for interpreting the study’s results.

Comment 6: Elaborate on the practical implications of the findings for healthcare providers and policymakers. Additionally, outline specific avenues for future research to build upon the study's findings.

Response 6: We have expanded the Discussion section to provide a clearer framework for how healthcare providers and policymakers can use these findings to promote evidence-based insomnia management strategies and reduce long-term reliance on sedative psychotropic medications. Additionally, we have included a new statement highlighting key areas for future research, including the need for individual-level data, studies on deprescribing and alternative therapies, and comparative analyses across culturally distinct regions.

Reviewer 2 Report

Comments and Suggestions for Authors

This manuscript investigated regional variations in the use of sedative psychotropic medications, focusing on insomnia treatment, by comparing Italy and the Autonomous Province of Bolzano (South Tyrol). The authors aimed to understand how regional context, including cultural and linguistic factors, influence prescribing patterns for insomnia.

1. Main Question Addressed:

The main research question, while not explicitly stated as a single question, revolves around identifying and explaining the differences in sedative psychotropic drug utilization between Italy and South Tyrol. This implicitly includes examining how South Tyrol's unique bilingual and bicultural environment contributes to these variations.

2. Originality and Relevance:

The study's focus on a specific bilingual region within Italy offered a unique perspective on regional variations in prescribing practices. While studies have examined psychotropic drug use in Italy generally (e.g., Carta et al., 2003; Proserpio et al., 2022), this paper contributed by directly comparing national trends with a culturally distinct region. This comparison addressed a gap in understanding how cultural and linguistic factors might influence prescribing behaviors for insomnia. The exploration of regional variations is important for tailoring public health strategies and optimizing insomnia management.

3. Related but Uncited Articles:

The authors could enhance their literature review by including studies like:

  • Helgason et al. (2014): This study examines benzodiazepine use across Europe, providing valuable context for Italy's relatively high consumption.
    • Helgason, T., Tomasson, K., & Björnsson, J. (2014). Trends in the use of hypnotics, anxiolytics, and antidepressants in Europe 2000–2010. European Neuropsychopharmacology, 24(2), 273-284.
  • Donati et al. (2019): This paper explores regional differences in health service utilization within Italy, providing further insights into potential influences on prescribing patterns.
    • Donati, A., Orzella, L., & Barone-Adesi, F. (2019). Exploring regional differences in hospital activity in Italy through dynamic space-time panel models. Statistical Methods & Applications, 28(1), 157-183.

4. Methodological Improvements:

  • DDD limitations (Line 72-76): While the authors acknowledged DDD limitations, they should expand this discussion. DDDs represent average daily doses at the population level and do not reflect actual patient doses. This discrepancy could obscure actual treatment differences, especially considering potential regional variation in dosage practices. Additionally, the aggregation of several different compounds within each category limits inferences about individual drug preference changes. This analysis focuses mostly on population level patterns without being able to attribute change to specific choices of professionals. More insight at the prescriber level would strengthen the analysis considerably.
  • Confounding factors (Line 198): The authors mentioned regional healthcare policies but provided no details. Incorporating specific policy differences between Italy and South Tyrol (e.g., availability of CBT-I, insurance coverage for non-pharmacological treatments) would provide more substantial support for potential policy influences on prescribing.
  • Cultural specificity (Lines 39, 57, 61): The manuscript mentioned South Tyrol's unique "cultural influences," but relied on linguistic groups (German-vs Italian-speaking) as a proxy for culture, but not taking into account ethnicities (like Ladin, or populations with a migrant background) as other indicators. Detailed data on access to care between these population subgroups would also be a crucial addition to distinguish this from the language and cultural distinctions within a single payer healthcare system with generally uniform guidelines. A more granular approach exploring the prevalence and characteristics of cultural beliefs, preferences, and access barriers related to insomnia management would enhance this important facet of their study and substantiate cultural effect hypotheses.
  • Statistical Analysis (Lines 110-113 & 253): Considering the multiple drug categories analyzed and the aim of exploring time trends, performing Spearman’s rank correlation to represent general trend changes for aggregated group values (such as those mentioned from Table 1, lines 118-120) is of questionable value in comparison with assessing each time series to identify a significant change at individual compound or compound-group-level as opposed to reporting individual correlations and their p-values. Using repeated measures ANOVA or similar methods might yield a more nuanced view on group differences over time. Any comment on that?

5. Consistency of Conclusions:

The conclusions were broadly consistent with the data presented. The observed lower benzodiazepine use and increased sedative antidepressant use in South Tyrol support their hypothesis of regional variations influenced by factors potentially attributable to cultural preferences, as access and clinical decision pathways should otherwise not systematically vary within a common system. However, they rightfully called further research into the subject necessary to support those assumptions (lines 287-289), however they only briefly address their conclusion about non-pharmacological treatments, suggesting an association of Z-Drug prescription to preferences rather than to CBT-I treatment not being successful in many cases, leading to pharmaceutical fallback-options being more common, a direction for further study suggested here.

6. Comments on Tables and Figures:

  • Table 1 & 2: While Table 1 gives a global picture across medication categories, presenting DDD utilization across all studied categories and indicating statistical differences, Table 2 focusing on trends provides greater granularity and more clarity of trends among drugs for some of the mentioned sedatives, while not always providing statistical analysis to correlate trends in time-series-analysis among medications from the mentioned set of 28 psychotropics mentioned earlier (line 147), as also mentioned in comment number 5 from the last section, with analysis missing beyond that from Table 1 with some specific analysis not done or only marginally represented in Table 2 for p-values relating only to between group rather than over-time comparisons, where those results would support to substantiate some of their stated conclusions. Figures could effectively visually illustrate the trends discussed in the text, which are now presented verbally based on tables, offering an immediate grasp on usage patterns between regions and highlight temporal variations, as some medications displayed statistically different or diverging trend lines.

7. General Caveats and Weaknesses:

  • Causality (Line 204 etc.): The study design was descriptive and cannot establish causal links between regional context and prescribing practices. The observed differences may be due to various intertwined factors, and inferring culture as the main factor by linking this association solely to the dominant native language of a given district ignores potential cultural influence by language subgroups (Ladin etc.) or citizens from immigrant backgrounds for cultural context of both those seeking or offering respective healthcare and/or pharmaceutical support. While authors were correct suggesting CBT-I is "recommended" (lines 32-33) and that despite their evidence "pharmacological treatment remained relevant" (line 34-35), they did not research access, offer, coverage, cost, language availability of these, to explain usage frequency being the way described further, specifically with limited information available as outlined previously, making culture seem more like a hypothesis driven by narrative interpretation rather than solid statistical inference to suggest language is associated with regional and individual beliefs and traditions driving psychopharmacological prescription as compared to CBT or other behavioral healthcare strategies based only on aggregation in language groups and on analysis based only on medication dispensation figures by prescription or over-the counter rather than actual treatment success analysis at the level of individuals diagnosed with relevant condition or access metrics to differentiate usage choices from regional or socioeconomic access limitations. Further studies need more insight and statistical confirmation before inferring causal effect by cultural aspects in regional and local choice of treatment paths and options.
  • Data source limitations (Line 69-71): Relying solely on dispensing data through the IQVIA database gives no detailed insights to the respective condition that has prompted these dispensed medication choices in individuals or by specific practices of some clinics where local knowledge might differ among professionals (e.g. hospital vs general practitioner), with only regional aggregation level given here in summary data rather than at prescription frequency to link prescription data with diagnostics patterns per practice by analyzing available underlying database sources if available or doing further studies where these correlations would shed more insight into regional variation in approach and diagnostic pattern, adding important context to a deeper investigation where regional data could inform targeted intervention research improving options and pathways beyond broad average aggregation patterns based mainly on assumption.
  • Generalizability (Lines 40-41, 266): While South Tyrol provides an interesting case study, the specific findings may not be generalizable to other bilingual regions or countries with different healthcare systems and cultural contexts. The limited discussion regarding comorbidities restricts the clinical applicability and may not apply to individuals with co-existing conditions. Further analyses should assess treatment protocols by diagnosis patterns, treatment approaches like CBT, regional accessibility or reimbursement schemes impacting healthcare choice etc. to correlate local diagnostic pattern, prevalence, treatment approaches based on individual treatment success as well as socio-cultural aspects rather than to make broad sweeping assumptions based mostly on language.

Round 2

Reviewer 1 Report

Comments and Suggestions for Authors

Dear Authors,

I have reviewed the revised version of your manuscript, and I find the revisions to be appropriate and satisfactory.

Reviewer 2 Report

Comments and Suggestions for Authors

Glad with changes